# Comparative Effects of GLP-1 and GLP-2 on Beta-Cell Function, Glucose Homeostasis and Appetite Regulation

**DOI:** 10.3390/biom14121520

**Published:** 2024-11-27

**Authors:** Asif Ali, Dawood Khan, Vaibhav Dubey, Andrei I. Tarasov, Peter R. Flatt, Nigel Irwin

**Affiliations:** Centre for Diabetes, School of Biomedical Sciences, Ulster University, Cromore Road, Coleraine BT52 1SA, Northern Ireland, UK; asif_ali-aa@ulster.ac.uk (A.A.); d.khan@ulster.ac.uk (D.K.); dubey-v@ulster.ac.uk (V.D.); a.tarasov@ulster.ac.uk (A.I.T.); pr.flatt@ulster.ac.uk (P.R.F.)

**Keywords:** appetite, beta-cell, cAMP, incretin, insulin secretion, islet

## Abstract

Glucagon-like peptide-1 (GLP-1) and glucagon-like peptide-2 (GLP-2) are related intestinal L-cell derived secretory products. GLP-1 has been extensively studied in terms of its influence on metabolism, but less attention has been devoted to GLP-2 in this regard. The current study compares the effects of these proglucagon-derived peptides on pancreatic beta-cell function, as well as on glucose tolerance and appetite. The insulin secretory effects of GLP-1 and GLP-2 (10^−12^–10^−6^ M) were investigated in BRIN-BD11 beta-cells as well as isolated mouse islets, with the impact of test peptides (10 nM) on real-time cytosolic cAMP levels further evaluated in mouse islets. The impact of both peptides (10^−8^–10^−6^ M) on beta-cell growth and survival was also studied in BRIN BD11 cells. Acute in vivo (peptides administered at 25 nmol/kg) glucose homeostatic and appetite suppressive actions were then examined in healthy mice. GLP-1, but not GLP-2, concentration dependently augmented insulin secretion from BRIN-BD11 cells, with similar observations made in isolated murine islets. In addition, GLP-1 substantially increased [cAMP]*_cyt_* in islet cells and was significantly more prominent than GLP-2 in this regard. Both GLP-1 and GLP-2 promoted beta-cell proliferation and protected against cytokine-induced apoptosis. In overnight fasted healthy mice, as well as mice trained to eat for 3 h per day, the administration of GLP-1 or GLP-2 suppressed appetite. When injected conjointly with glucose, both peptides improved glucose disposal, which was associated with enhanced glucose-stimulated insulin secretion by GLP-1, but not GLP-2. To conclude, the impact of GLP-1 and GLP-2 on insulin secretion is divergent, but the effects of beta-cell signaling and overall health are similar. Moreover, the peripheral administration of either hormone in rodents results in comparable positive effects on blood glucose levels and appetite.

## 1. Introduction

Glucagon-like peptide-1 (GLP-1) and glucagon-like peptide-2 (GLP-2) are peptide hormones derived from the same precursor gene, known as proglucagon, and co-secreted in equimolar concentrations by enteroendocrine L-cells in response to nutrient ingestion [1]. The proglucagon gene product undergoes tissue-specific post-translational processing to yield different bioactive peptides, including GLP-1 and GLP-2 [2]. Thus, in the intestine and brain, proglucagon is processed by prohormone convertase 1/3 (PC1/3) at Arg-Arg sites to yield glicentin, oxyntomodulin (OXM), intervening peptide-2 (IP-2), GLP-1 and GLP-2. Conversely, in alpha-cells of the endocrine pancreas, proglucagon is cleaved by PC2 to generate glicentin-related pancreatic polypeptide (GRPP) intervening peptide-1 (IP-1), the major proglucagon fragment (MPGF) and glucagon [3,4]. Whilst there is evidence of islet synthesis and the secretion of GLP-1 and GLP-2 under conditions of islet stress [5,6], these hormones are still largely considered as intestinal-derived peptides.

To date, there is a plethora of literature relating to the metabolic benefits of GLP-1 receptor activation, highlighted by the clinical approval of GLP-1 mimetics for both type 2 diabetes and obesity [7,8]. Thus, the positive modulation of GLP-1 receptor signaling leads to an enhancement of glucose-stimulated insulin secretion (GSIS) from pancreatic beta-cells, together with the inhibition of glucagon secretion from alpha-cells [7], the slowing of gastric emptying and the promotion of satiety [2]. On the other hand, GLP-2 receptor activation has been shown to exert benefits on intestinal growth and repair through the promotion of epithelial proliferation [9], leading to the clinical application of GLP-2 drugs for short bowel syndrome [10]. Despite the structural similarities of GLP-1 and GLP-2, alongside the fact that both hormones are secreted in response to nutrient ingestion [1], there is a lack of information about the possible metabolic benefits of GLP-2. However, GLP-2 may have the potential to protect against the dysregulation of glucose metabolism, as well as positively modulate energy homeostasis [11,12,13].

Therefore, the present study directly compares the impact of GLP-1 and GLP-2 on pancreatic BRIN BD11 beta-cell proliferation, survival and overall secretory function. The effects of both peptides on insulin secretion were then verified in isolated murine islets, along with their influence on islet cell cytosolic cAMP concentrations. In addition, we investigated the effects of the intraperitoneal injection of GLP-1 and GLP-2 on glucose tolerance and GSIS in mice. Finally, the influence of both peptides on appetite regulation in mice fasted overnight, as well as mice trained to eat for 3 h per day, was examined. Overall, the data further emphasize the prominent anti-diabetic and -obesity effects of GLP-1 receptor signaling. Furthermore, we also reveal, for the first time, the positive impact of GLP-2 receptor signaling on islet cell cAMP levels, as well as beta-cell turnover, meriting further investigation in terms of therapeutic strategies for diabetes.

## 2. Materials and Methods

### 2.1. Peptides

Peptides (95% purity) were obtained from a commercial source (Synpeptide, Shanghai, China) and fully characterized in our laboratory, as previously described [14].

### 2.2. In Vitro Insulin Secretion

The BRIN-BD11 beta-cell line [15] was used to examine the insulin secretory actions of the test peptides (*n* = 8; 20 min incubation; 10^−12^–10^−6^ M) at 5.6 and 16.7 mM of glucose, as described previously [16]. Furthermore, the impact of the peptides on insulin secretion (*n* = 4; 60 min incubation; 10^−8^ and 10^−6^ M) was also examined in an islet isolated from 12-week-old C57BL/6 mice by collagenase digestion [16]. The subsequent acid–ethanol extraction of test islets allowed for the presentation of islet secretion data as a percentage of the islet insulin content. Samples were kept at −20 °C prior to insulin determination using an in-house radioimmunoassay [17].

### 2.3. Live Islet Cell Time-Lapse Imaging

C57BL/6 male mouse (12 weeks old) islets were isolated as above. To quantify the cytosolic cAMP levels, a recombinant fluorescent sensor (Upward Green cADDis, Montana Molecular, Bozeman, MT, USA) was used. The sensor was delivered to the islets via adenoviral transduction, allowing 24–48 h for gene expression. Time-lapse imaging of [cAMP]***_cyt_*** was performed using the Green Upward cADDis sensor, as described previously [18], with image acquisition managed using μManager 2.0 software, capturing the cAMP levels in the islets every 60 s (16 mHz). For imaging, a bath perfusion solution (140 mM of NaCl, 4.6 mM of KCl, 2.6 mM of CaCl_2_, 1.2 mM of MgCl_2_, 1 mM of NaH_2_PO_4_, 5 mM of NaHCO_3_, 10 mM of glucose, 10 mM of HEPES, pH 7.4) was used, along with GLP-1, GLP-2 (both at 10 nM) or IBMX (100 µM) as a positive control. Image sequences were analyzed with the use of the open-source FIJI software version 2.9.0 (National Institutes of Health (NIH), Bethesda, MD, USA).

### 2.4. Beta-Cell Proliferation and Cellular Stress Studies

The impact of GLP-1 and GLP-2 on BRIN-BD11 beta-cell proliferation (40,000 cells per well) was assessed using the Ki67 primary antibody (Ab15580, AbCam, Cambridge, UK) and the Alexa Fluor^®^ 594 secondary antibody, as described previously in our laboratory [19]. For apoptosis studies, cellular stress was induced by the incubation of BRIN BD11 beta-cells with a cytokine cocktail (IL-1β 100 U/mL, IFN-γ 20 U/mL, TNF-α 200 U/mL), and the rate of apoptosis was monitored through TUNEL staining (Fluorescein, Roche Diagnostics, Burgess Hill, UK) [19]. The effects were visualized using a fluorescence microscope (Olympus system microscope, model BX51; Southend-on-Sea, UK) and a DP70 camera adapter system using DAPI (350 nm), TRITC (594 nm) and FITC (488 nm) filters, alongside an Olympus XM10 camera. For quantification, the cell-counter function within ImageJ Software Version 1.54 (National Institutes of Health (NIH), Bethesda, MD, USA) was employed to establish the number of positively stained Ki-67 or TUNEL cells, as appropriate; the data were then presented as a percentage of the total cells investigated.

### 2.5. Animal Experiments

Animal experiments were performed in male C57BL/6 (12–14 weeks of age) or NIH Swiss male mice (30 weeks of age), as appropriate; both were purchased from Harlan Ltd., Huntingdon, UK. Animals were housed individually in the Biomedical and Behavioural Research Unit (BBRU) at Ulster University for pre-clinical studies, with a standard temperature and light cycle, namely 22 ± 2 °C and a 12 h light/dark cycle, respectively. All procedures were performed in compliance with the UK Animal Scientific Procedures Act 1986.

### 2.6. In Vivo Experiments

The effects of GLP-1 or GLP-2 (25 nmol/kg bw, an intraperitoneal (i.p.) administration) on food intake, glucose homeostasis and insulin secretion were studied in overnight fasted C57BL/6 mice, as previously documented in our laboratory [19]. The dosing regimen employed for the test peptides was based on previous positive observations with GLP-1 and GLP-2, as well as related gut-derived peptide hormones, within the same experimental systems [19]. In a separate series, NIH male mice habituated to a daily feeding regime of 3 h/day were also used to evaluate the impact of GLP-1 and GLP-2 (25 nmol/kg bw) on the cumulative food intake, using the same protocol as described above. These mice were subject to a progressive reduction in the daily feeding period over 3 weeks at 12 weeks of age, as detailed previously [20]. Mice were maintained on this 3 h/day feeding regimen until 30 weeks of age, and experiments were then conducted.

### 2.7. Biochemical Analyses

Blood glucose levels were quantified using a blood glucose meter (Ascencia Contour; Bayer Healthcare, Berkshire, UK). For plasma insulin analysis, blood samples were collected into chilled fluoride/heparin glucose micro-centrifuge tubes (Sarstedt, Numbrecht, Germany), immediately centrifuged at 13,000× *g* for 1 min, and retained at −20 °C before insulin quantification by radioimmunoassay [17].

### 2.8. Statistical Analyses

Statistical analyses were conducted using GraphPad PRISM software (Version 8.0, Irvine, CA, USA). Data are presented as mean ± S.E.M. Comparative analyses between groups were performed using one-way ANOVA with a Bonferroni post hoc test or Student’s unpaired *t*-test, as appropriate. Statistical significance was defined as *p* < 0.05.

## 3. Results

### 3.1. GLP-1, but Not GLP-2, Evokes Prominent Insulin Secretion from BRIN BD11 Beta-Cells and Isolated Islets

At 5.6 and 16.7 mM of glucose, GLP-1 (10⁻^10^–10⁻^6^ M) significantly (*p* < 0.05–0.001) augmented insulin secretion when compared to the controls (Figure 1A,B). Conversely, GLP-2 did not impact insulin release from BRIN BD11 cells at any of the concentrations tested at either 5.6 or 16.7 mM glucose (Figure 1A,B). Equivalent observations were made at 16.7 mM of glucose in collagenase-isolated mouse islets (Figure 1C).

### 3.2. GLP-1 and GLP-2 Elevate Cytosolic cAMP Concentrations in Isolated Mouse Islets

The incubation of murine islets with 10 mM of glucose in combination with 10 nM of GLP-1 resulted in a rapid and sustained increase (*p* < 0.01) in [cAMP]*_cyt_* (Figure 2A,B). GLP-2 also enhanced (*p* < 0.01) [cAMP]*_cyt_* in mouse islet cells, but was significantly less effective (*p* < 0.01) than GLP-1 (Figure 2A,B). As expected, the incubation of mouse islet cells with 100 µM of IBMX led to prominent increases (*p* < 0.01) in [cAMP]*_cyt_* (Figure 2A,B). In agreement, the impact of GLP-1 and IBMX on islet cell [cAMP]*_cyt_* was moderately positively correlated (r = 0.56), whereas similar correlations for GLP-2 with either IBMX (r = −0.18) or GLP-1 (r = −0.08) were not observed (Figure 2C).

### 3.3. GLP-1 and GLP-2 Promote BRIN BD11 Beta-Cell Proliferation and Protect Against Cytokine-Induced Apoptosis

Both GLP-1 and GLP-2 (10⁻^8^ and 10⁻^6^ M) significantly increased (*p* < 0.001) BRIN BD11 cell proliferation following overnight culture (Figure 3A). Representative images of Ki-67-stained cells are also presented, with arrows indicating proliferating cells (Figure 3B). Similar to observations regarding proliferation, both GLP-1 and GLP-2 (10⁻^8^ and 10⁻^6^ M) significantly enhanced (*p* < 0.001) beta-cell survival and protected against cytokine-induced beta-cell apoptosis (Figure 4A). Representative images of TUNEL-stained BRIN BD11 cells are provided, with arrows indicating apoptotic cells (Figure 4B).

### 3.4. GLP-1 and GLP-2 Improve Glucose Tolerance and Supress Appetite in Mice

The I.p. administration of GLP-1 or GLP-2 conjointly with glucose, at a dose of 25 nmol/kg, decreased individual as well as overall glucose values (Figure 5A,B). Specifically, GLP-1 reduced blood glucose levels at 60 (*p* < 0.05) and 90 (*p* < 0.01) minutes post-injection, with GLP-2 significantly reducing (*p* < 0.05) glucose only at 90 min (Figure 5A). Accordingly, the 0–90 min overall AUC glucose values were decreased by both GLP-1 (*p* < 0.01) and GLP-2 (*p* < 0.05) administration (Figure 5B). As expected, GLP-1 also augmented GSIS, corresponding to increased (*p* < 0.05) plasma insulin concentrations at 15 min post-injection, as well the overall values at 0–90 min (*p* < 0.01) (Figure 5C,D). In contrast, the effects of GLP-2 on insulin secretion failed to reach significance (Figure 5C,D). In terms of appetite suppression, both GLP-1 and GLP-2 reduced food intake (*p* < 0.05–0.001) at all observation points in the overnight fasted mice (Figure 6A). Both peptides were found to have similar significant (*p* < 0.05–0.01) appetite suppressive effects in mice trained to eat for 3 h/day, although GLP-2 lost efficacy in this model at 180 min post-injection (Figure 6B).

## 4. Discussion

In recent years, the products of the proglucagon gene have generated significant interest in the context of the regulation of glucose homeostasis and body composition. Among these, GLP-1 has emerged as a central player, with established therapeutic applicability for type 2 diabetes and obesity [2]. GLP-2 is a related proglucagon gene product that has structural and amino acid sequence homology with GLP-1 and has been found to promote intestinal growth and integrity, but has less well characterized effects on metabolism [21]. However, similar to GLP-1, GLP-2 receptor expression is evidenced in both human and rodent pancreatic islets, suggesting that it plays a direct role in islet function and overall metabolism [19,22]. Moreover, the GLP-2-mediated suppression of appetite has recently been observed in rodents and been demonstrated to be dependent on hypothalamic GLP-1 receptor signalling [23]. Thus, GLP-2 may have an overlapping bioactivity profile with GLP-1 that merits further consideration.

As anticipated, GLP-1 induced prominent concentration-dependent insulin secretory actions in BRIN BD11 beta-cells, as well as isolated mouse islets, that were associated with increased intracellular cAMP [24]. More intriguingly, we were also able to evidence small, but significant, GLP-2-mediated elevations in islet cell cAMP concentrations. In this regard, we have previously shown that GLP-2 does not alter the membrane potential or intracellular Ca^2+^ levels in pancreatic beta-cells [19]; thus, GLP-2 likely signals via cAMP and related downstream effectors such as protein kinase A (PKA) or exchange protein activated by cAMP (Epac) in islets, similar to GLP-1 [25]. This corresponds well with that observation that GLP-2 enhanced BRIN BD11 cell proliferation, as well as protection against cytokine-induced apoptosis, in the current study [26]. Such findings are also consistent with the well-documented proliferative actions of GLP-2 within the intestine [9]. That said, and in good agreement with others [6,19], GLP-2 did not augment GSIS within either the in vitro, ex vivo or in vivo environments. Similar differential effects of compounds on beta-cell function have been reported previously [27], as well as with agents such as Peptide YY (PYY) and Pancreatic Polypeptide (PP), which can inhibit insulin secretion but augment beta-cell growth and survival [4].

To add to the complexity, GLP-2 has been demonstrated to function as a low potent agonist of the GIP receptor [28], with GIP being well known to augment beta-cell function [29]. Thus, further investigations such as islet-cell-specific receptor knockdown studies would be required to fully delineate the effects of GLP-2 at the level of the beta-cell. This is even more relevant given the observations of local pancreatic islet GLP-2 synthesis and secretion in both rodent and human islets [6,19]. In addition, GLP-2 receptor expression is considered to be relatively low on beta-cells [6]. Nonetheless, the equivalent effects of GLP-2 and GLP-1 on beta-cell turnover are intriguing and merit further investigation given that all types of diabetes are ultimately linked to the loss of beta-cell mass and function [30]. The anti-apoptotic role of GLP-1 in various cell types is already well documented [2], with GLP-2 previously demonstrated to protect against streptozotocin-induced DNA damage in beta-cells [19], as well as dextran sulphate-induced colitis and obesity-related neuroinflammation in mice [31,32]. Overall, it appears that GLP-2 possesses beta-cell-sparing activities that could be highly relevant for diseases such as diabetes.

Interestingly, both GLP-1 and GLP-2 improved glucose handling following combined injection with glucose in mice. For GLP-1, this can largely be attributed to GSIS [33], whereas for GLP-2, improvements in insulin action and/or insulin-independent glucose uptake would seem the most likely explanation. In this respect, there is an inverse correlation between the GLP-2 concentrations and insulin sensitivity in obese human subjects [34]. Moreover, GLP-2 receptor activation has been shown to prevent the glucose dysregulation that occurs following the induction of insulin resistance by prolonged high-fat feeding in mice [13], partly through enhancing insulin signalling [35]. In agreement with others [12,23], GLP-2 curbed food intake in mice to a similar degree as GLP-1, consistent with the knowledge that GLP-2 can directly regulate the hypothalamic neurons linked to appetite control [36]. However, unlike GLP-1 [37,38], the full translation of the appetite suppressive actions of GLP-2 is yet to be confirmed. However, our reliable observations in a well characterized rat beta-cell line and mouse model, alongside knowledge that the proglucagon gene is highly conserved across mammalian species [39], would suggest good translatability. Overall, GLP-2 exerts benefits on metabolism that are akin to those of GLP-1 [2], likely mediated through distinct pathways. It follows that a unimolecular dual GLP-1/GLP-2 receptor agonist, known as GUB09-145, suppressed caloric intake, promoted weight loss, and improved glucose tolerance in obese mice [40]. Furthermore, another recently characterized GLP-1/GLP-2 hybrid peptide, namely PG-102, was demonstrated to exert benefits on beta-cell mass and glucose control in *ob*/*ob* mice that were superior to either tirzepatide or retatrutide [41]. Thus, although the pancreatic beta-cell secretory actions of GLP-1 and GLP-2 are distinct, parallels in terms of their beneficial effects on beta-cell signalling and turnover, as well as glucose homeostasis and appetite regulation, provide an attractive avenue for therapeutic application. That said, there is some evidence that GLP-2 could accelerate the growth of colonic neoplasms [42]; thus, the potential intestinotrophic effects of GLP-2 would need to be appropriately moderated within this setting.

## 5. Conclusions

Interest in GLP-1 has increased significantly within the clinical field of diabetes and obesity. However, other related proglucagon-derived peptides, such as GLP-2, also merit attention. As such, despite GLP-2 lacking direct beta-cell secretory actions, this hormone supports overall beta-cell health by encouraging growth and preventing the destruction of these cells. In addition, we have shown, for the first time, that GLP-2 positively impacts cAMP oscillations in islet cells, giving credence to the notion that locally produced GLP-2 mediates the important crosstalk between endocrine cells [22]. When viewed alongside satiety and glucose homeostatic actions, this suggests that GLP-2 has untapped potential for the treatment of diabetes and obesity, either alone or perhaps more likely alongside GLP-1 therapies.

## Figures and Tables

**Figure 1 biomolecules-14-01520-f001:**
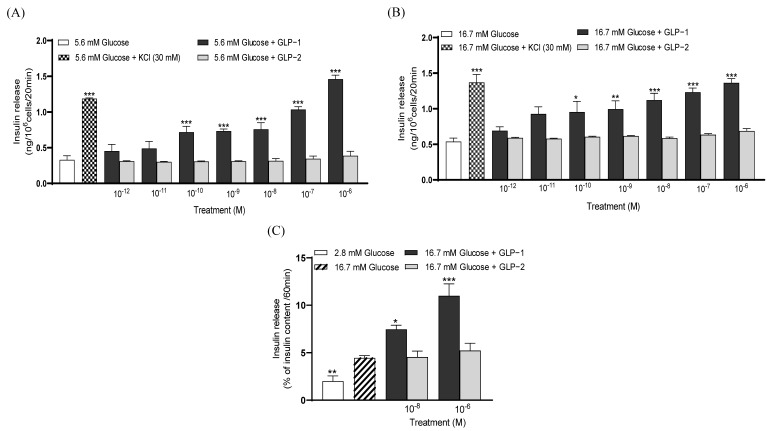
Effects of GLP-1 and GLP-2 on insulin secretion from (**A**,**B**) BRIN BD11 beta and (**C**) isolated mouse islets. BRIN BD11 cells were incubated for 20 min with (**A**) 5.6 or (**B**) 16.7 mM of glucose alone and in combination with test peptides (10^−12^ to 10^−6^ M), and the insulin secretion determined. (**C**) Isolated mouse islets were incubated for 60 min with peptides (10^−8^ and 10^−6^ M) at 16.7 mM of glucose (10 islets per well), and the insulin secretion was determined. Values are mean ± SEM (*n* = 8). * *p* < 0.05, ** *p* < 0.01, *** *p* < 0.001, compared to the respective glucose control.

**Figure 2 biomolecules-14-01520-f002:**
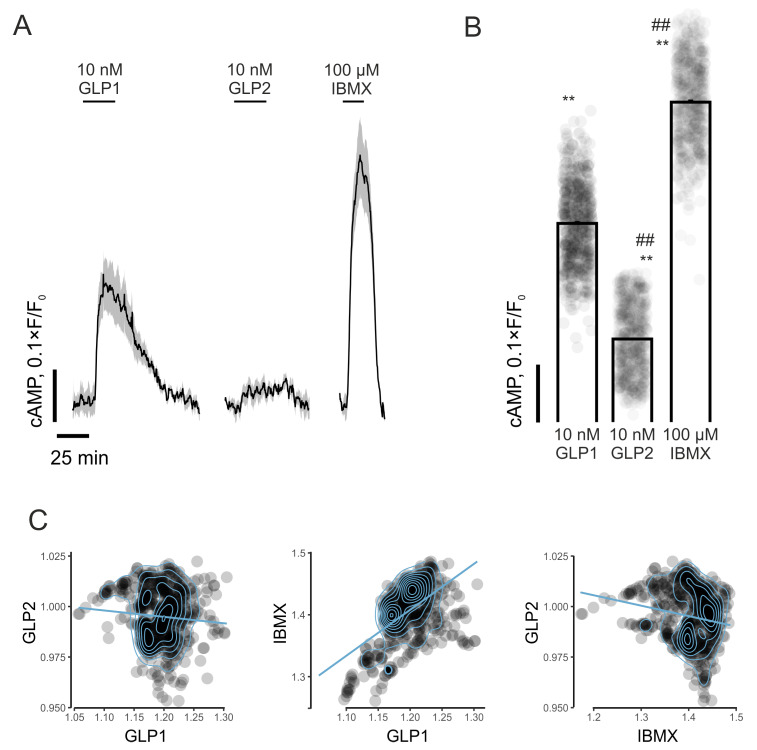
Effects of GLP-1 and GLP-2 on real-time cytosolic cAMP concentrations in isolated mouse islets. (**A**) Dynamics of [cAMP]***_cyt_*** in mouse islet cells in response to incubation with GLP-1, GLP-2 (both at 10 nM) or IBMX (100 µM) in the presence of 10 mM of glucose. (**B**) Quantification of islet cell [cAMP]***_cyt_*** responses to GLP-1, GLP-2 (both at 10 nM) or IBMX (100 µM). (**C**) Pearson correlation (per-cell) between [cAMP]***_cyt_*** responses for GLP-1, GLP-2 and IBMX in islets. Values are mean ± SEM (*n* = 615 from three preparations). ** *p* < 0.01 compared to basal condition. ^##^ *p* < 0.01 compared to 10 nM of GLP-1.

**Figure 3 biomolecules-14-01520-f003:**
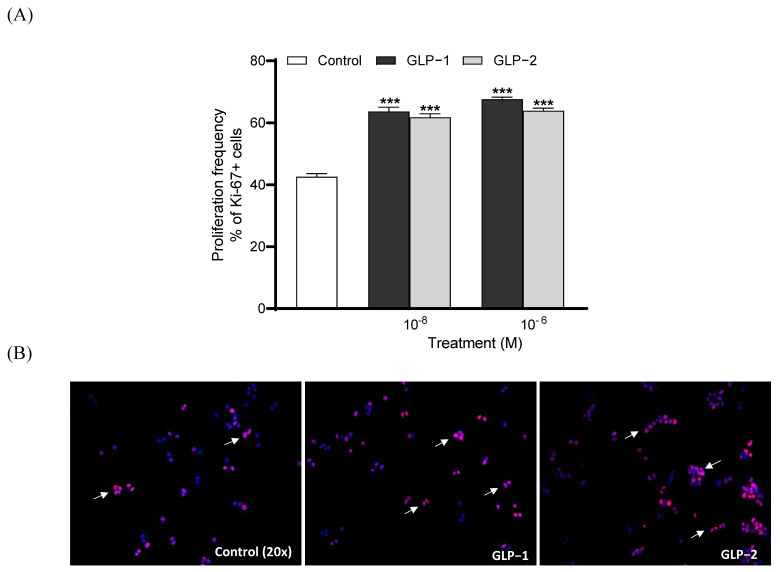
Effects of GLP-1 and GLP-2 on BRIN-BD11 cell proliferation. BRIN-BD11 cells were incubated with GLP-1 or GLP-2 (10^−6^ M and 10^−8^ M) for 18 h. Cells were then stained for Ki-67 (red) and DAPI (blue). The quantification of Ki-67 positive cells is shown in (**A**). Representative images (**B**) show Ki-67-positive cells indicated by arrows. Values are expressed as mean ± SEM (*n* = 6). *** *p* < 0.001 compared to media control.

**Figure 4 biomolecules-14-01520-f004:**
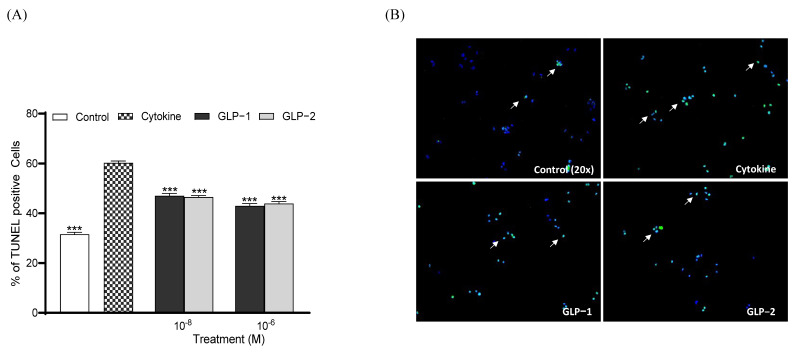
Effects of GLP-1 and GLP-2 on BRIN-BD11 cell apoptosis. BRIN BD11 cells were incubated in the presence of cytokines (IL-1β 100 U/mL, IFN-γ 20 U/mL, TNF-α 200 U/mL) alone or alongside GLP-1 or GLP-2 (10^−6^ M and 10^−8^ M) for 18 h before staining for TUNEL (green) or DAPI (blue). The quantification of TUNEL positive cells is shown in (**A**). Representative images (**B**) show TUNEL-positive cells indicated by arrows. Values are expressed as mean ± SEM (*n* = 4). *** *p* < 0.001 compared to the cytokine group.

**Figure 5 biomolecules-14-01520-f005:**
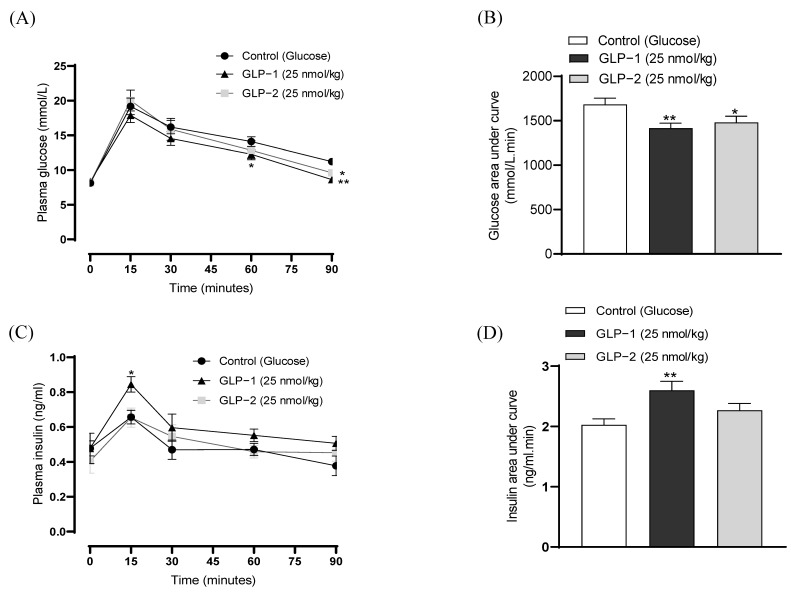
Effects of GLP-1 and GLP-2 on glucose tolerance and insulin secretion in mice. (**A**) Blood glucose and (**C**) plasma insulin was assessed following the administration of glucose alone (18 mmol/kg bw) or in combination with GLP-1 and GLP-2 (25 nmol/kg bw). (**B**,**D**) Overall 0–90 min AUC (**B**) glucose and (**D**) insulin values are also shown. Values are mean ± SEM (*n* = 6). * *p* < 0.05, ** *p* < 0.01, compared to the glucose control.

**Figure 6 biomolecules-14-01520-f006:**
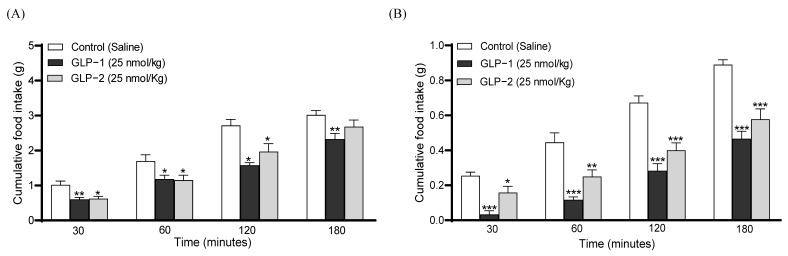
Effects of GLP-1 and GLP-2 on food intake in mice. Saline vehicle (0.9% NaCl), GLP-1 or GLP-2 (25 nmol/kg bw, i.p.) were administered to (**A**) overnight fasted mice or (**B**) mice trained to eat for 3 h per day, and the cumulative food intake was recorded at 30 min intervals for 3 h. Values are mean ± SEM (*n* = 6). * *p* < 0.05, ** *p* < 0.01, *** *p* < 0.001, compared to the respective saline control.

## Data Availability

The authors declare that the data supporting the findings of this study are available within the article. Any additional raw data supporting the conclusions of this article will be made available by the senior author (N.I.), without undue reservation.

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
