# Peer review of "Comparative Effects of GLP-1 and GLP-2 on Beta-Cell Function, Glucose Homeostasis and Appetite Regulation"

_biomolecules, 2024, doi:10.3390/biom14121520_

Round 1

Reviewer 1 Report

Comments and Suggestions for Authors

This manuscript reports an original study to compare the metabolic effects of GLP-1 and GLP-2 using a cultured insulin-secreting cell line, isolated pancreatic islets and in vivo procedures in mice.  The main findings are that GLP-2 (unlike GLP-1) does not acutely enhance insulin secretion (partly explained by differences of intracellular cAMP responses), whereas both hormones similarly increase islet beta-cell proliferation and both confer a degree of protection against cytokine-induced apoptosis.  Also, both hormones improve glucose tolerance and reduce appetite, although these effects are less potent with GLP-2.   The study uses an appropriate range of experimental approaches that are well established by this group, and the findings are clearly presented and discussed in a balanced manner. 

Methods, sect 2.6, L136+. The authors may wish to mention why a dose of 25 nmol/kg body weight ip was selected.

Discussion end L311.   Given the intestinal effects of GLP-2, and experience with use in short bowel syndrome and on colonic neoplasms, should the potential to provide an attractive avenue for therapeutic application in diabetes be moderated with a note of caution – if intestinotropic effects are appropriately contained.

Fig 3 images or legend (B) require confirmation of control/GLP-1/GLP-2.  Similar for images in Fig 4.

Typographical. L41, ‘Whereas’ not fit easily as start of new sentence: suggest ‘conversely’.  L187, leads. L288 remove underlines.

Author Response

Reviewer 1:

We thank the Reviewer for their positive assessment of our manuscript, in particular his/her comments that ‘The study uses an appropriate range of experimental approaches that are well established by this group, and the findings are clearly presented and discussed in a balanced manner’. Our responses to the minor comments raised by the Reviewer are considered below.

Specific comments

  1. The Reviewer suggests ‘Methods, sect 2.6, L136+. The authors may wish to mention why a dose of 25 nmol/kg body weight ip was selected’. The authors thank the Reviewer for the opportunity to clarify this matter. Our peptide dosing regimen was based on previous positive observations with GLP-1, GLP-2 and other related gut derived hormones within the same experimental systems. To highlight this within the revised manuscript, the following sentence has been added to Section 2.6 of the Methods section (lines 131-134), that reads as follows: ‘The peptide dosing regimen was based on previous positive observations with GLP-1 and GLP-2, as well as related gut-derived peptide hormones, on food intake and glucose homeostasis in rodent models [19]’.

  1. The Reviewer notes ‘Discussion end L311. Given the intestinal effects of GLP-2, and experience with use in short bowel syndrome and on colonic neoplasms, should the potential to provide an attractive avenue for therapeutic application in diabetes be moderated with a note of caution – if intestinotropic effects are appropriately contained’. We thank the Reviewer for this comment and agree with his/her suggestion. The following sentence has now been added to the revised Discussion section (lines 300-302), that reads as follows: ‘That said, there is some evidence for GLP-2 to accelerate the growth of colonic neoplasms [42], thus potential intestinotrophic effects of GLP-2 would need to be appropriately moderated within this setting’. The reference section has been updated accordingly.

  1. The Reviewer notes ‘Fig 3 images or legend (B) require confirmation of control/GLP-1/GLP-2. Similar for images in Fig 4’. We apologise for this administrative oversight and appropriate labels have now been added to the representative images in Figures 3 and 4. The authors thank the Reviewer for the opportunity to improve the transparency of our paper.

  1. The Reviewer highlights a small number of typographical/expression errors. We have updated the manuscript as suggested and are thankful for the chance to improve the quality of our paper.

Reviewer 2 Report

Comments and Suggestions for Authors

The manuscript by Asif, A. et al studied the effects and differences of GLP-1 and GLP-2 on beta cell function, glucose homeostasis and regulation of appetite.

The rationale to investigate their differences was based on the knowledge that they derive from the same precursor gene pro-glucagon and that they are secreted by the same entero-endocrine L-cells in response to nutrient intake. After post-translational processing they give rise to two bioactive peptides: GLP-1 and GLP-2.  Another interesting fact is that in the islets of Langerhans there is also synthesis and secretion of both peptides under conditions of stress. Although GLP-2 expression in islets is low.

Despite their structural similarities and that both respond to nutrient ingestion there is no information about the metabolic effects of GLP-2. This study sought to examine it.

They compared and examined their function on two in vitro experiments using BRIN-BD11and isolated murine islets also a vivo experiments testing overnight fasted healthy and mice trained to eat 3 hours per day.

The results showed that GLP-2 did not alter insulin release from BRIN-BD11 cells at normal and high concentrations of glucose, whereas GLP-1 did.

GLP-1 and GLP-2 elevated cytosolic concentrations of cAMP in isolated mouse islets by combination of exposure to high glucose and IBMX.

GLP-2 suppressed food intake similar to GLP-1 but the full translation of its appetite suppressive actions still needs to be confirmed. GLP-2 lacks direct secretory actions but supports growth and prevent destruction by apoptosis of beta cells.

The experimental procedures were well described, and the data obtained showed that GLP-1and GLP-2 secretory actions are distinct. They have parallel effects on beta cell signaling and cellular turnover and in glucose homeostasis and appetite regulation.

Comments:

1-       In the beta cell proliferation and cellular stress studies how was proliferation and apoptosis quantified. Missing in the materials and methods.

2-       If the cell line BRIN-BD11 is derived from rat, why were the in vitro and in vivo experiments conducted in mice?

Overall is an interesting well written manuscript with very minor comments.   

Author Response

Reviewer 2:

The authors thank the Reviewer for their positive and helpful evaluation of our manuscript, as well as his/her comments that ‘The experimental procedures were well described …’ and ‘Overall is an interesting well written manuscript with very minor comments’. The Reviewer has highlighted two minor points that are considered below.

Specific comments

  1. The Reviewer comments ‘In the beta cell proliferation and cellular stress studies how was proliferation and apoptosis quantified. Missing in the materials and methods’. We are happy to add this additional information as suggested. As such, the following sentence is now included within the Methods section (Section 2.4, lines 113-116), that reads: ‘For quantification, the cell-counter function within ImageJ Software was employed to establish numbers of positively stained Ki-67 or TUNEL cells, as appropriate, and then expressed as percentage of the total number of cells analyzed’.

  1. The Reviewer remarks ‘If the cell line BRIN-BD11 is derived from rat, why were the in vitro and in vivo experiments conducted in mice?’. We thank the Reviewer for the opportunity to address this matter. The proglucagon gene is highly conserved across mammalian species, with the biological actions of GLP-1 and GLP-2 fully consistent between humans and rodents. Thus, use of a well characterised rat beta-cell line alongside a mouse model helps to confirm translatability of our findings. To highlight this within the revised text, the following sentence has now been added to the Discussion section (lines 288-290), that reads as follows: ‘Although, our reliable observations in a well characterized rat beta-cell line and mouse model, alongside knowledge the proglucagon gene is highly conserved across mammalian species [39], would suggest good translatability. Overall, …..’. The reference section has been updated accordingly.